# Twisted Fiber Optic SPR Sensor for GDF11 Concentration Detection

**DOI:** 10.3390/mi13111914

**Published:** 2022-11-05

**Authors:** Yong Wei, Ze Ran, Rui Wang, Zhuo Ren, Chun-Lan Liu, Chun-Biao Liu, Chen Shi, Chen Wang, Yong-Hui Zhang

**Affiliations:** 1College of Electronic & Information Engineering, Chongqing Three Gorges University, Chongqing 404100, China; 2Chongqing Key Laboratory of Development and Utilization of Genuine Medicinal Materials in Three Gorges Reservoir Area, Chongqing Three Gorges Medical College, Chongqing 404120, China

**Keywords:** GDF11 concentration detection, twisted fiber optic SPR sensor, SPR biosensor, growth differentiation factor specific detection

## Abstract

There are few methods and insufficient accuracy for growth differentiation factor 11 (GDF11) concentration detection. In this paper, we designed a twisted fiber cladding surface plasmon resonance (SPR) sensor, which can achieve a high precision detection of GDF11 concentration. The new structure of the fiber cladding SPR sensor was realized by coupling the light in the fiber core to the cladding through fiber thermal fusion twisting micromachining technology; a series of functionalized modifications were made to the sensor surface to obtain a fiber sensor capable of GDF11 specific recognition. The experimental results showed when GDF11 antigen concentration was 1 pg/mL–10 ng/mL, the sensor had a detection sensitivity of 2.518 nm/lgC, a detection limit of 0.34 pg/mL, and a good log-linear relationship. The sensor is expected to play a role in the rapid and accurate concentration detection of pathological study for growth differentiation factors.

## 1. Introduction

Growth differentiation factor 11 (GDF11), as a member of transforming growth factor β (TGFβ) protein, is involved in many physiological and pathological processes. However, the studies on GDF11 function by different teams have obtained contradictory results. On the one hand, GDF11 is reported to be involved in the regulation of the aging process in the body and plays a protective role in the function of the brain, skeletal muscles, and heart [1,2]. Researchers at the Harvard Stem Cell Institute found that GDF11 levels decreased with age and that GDF11 reversed aging-related cardiovascular remodeling, muscle aging, and neural senescence [3]. However, studies from other laboratories have reached the opposite conclusion that GDF11 levels increased with age and GDF11 inhibited skeletal muscle regeneration [4]. At the same time, contradictory findings have been seen in studies of GDF11 on tumors [5]. One of the important reasons is the lack of uniformity in the detection standards and the lack of precision of the assay of GDF11 concentration by each team. Therefore, an accurate and rapid detection method of GDF11 concentration is important in the study of GDF11 in physiological and pathological processes.

In past studies, less work has been reported on the GDF11 concentration detection method. There are two main methods to perform GDF11 concentration assays. The first method is the enzyme-linked immunosorbent assay (ELISA). The antibody is labeled by the combination of the enzyme molecule and the antibody molecule. The labeled antibody is specifically combined with the antigen. After the substrate solution is dropped, the substrate will undergo a color reaction under the action of the enzyme. The color reaction can be analyzed by the ELISA detector [6]. However, this assay has strict requirements for the activity of the enzyme molecules to be labeled, which has an important impact on the accuracy of the detection of the concentration of the analyte to be measured. In addition, the interpretation of the chromogenic reaction by ELISA assay is limited by the accuracy of the instrument, which can also have an impact on the determination of GDF11 concentration. The second method is fluorescence immunoassay. The antibody (or antigen) is labeled with fluorescent materials and, after specific binding with the antigen (or antibody), fluorescence photos are taken using a fluorescence microscope. The properties, localization, and concentration of the antigen or antibody are analyzed by setting appropriate measurement indicators [7]. However, the strength of the fluorescence signal, poor labeling, and high background noise can directly affect the interpretation of the analysis results and inaccurate calibration of the concentration of the analyte to be measured. Therefore, it becomes an urgent problem to develop a new detection method for the direct, accurate, and real-time detection of GDF11 concentrations.

Optical fiber surface plasmon resonance (SPR) sensing technology, which has the advantages of non-labeling test samples, real-time online detection, and high sensing sensitivity, is widely used in biochemical sensing [8,9,10]. The detection principle is that the specific bindings of antigen and antibody molecules on the sensor surface change the refractive index of the external environment of the gold film on the sensor surface. The SPR phenomenon is very sensitive to the changes in the refractive index of the environment, which causes the SPR resonance valley to shift. Therefore, the movement of the SPR resonance valley can directly and sensitively reflect the change of the test sample concentration.

In this paper, a novel fiber optic cladding SPR sensor was constructed to achieve rapid detection of GDF11 concentration. The key problem for cladding the SPR sensor was effectively coupling the light in the fiber core to the cladding. It was solved by using the fiber hot-melt torsion micromachining technology. The cladding surface was plated with 50nm gold film, which constitutes the cladding SPR sensing structure that is easy to be used for biochemical detection. The antigen–antibody of GDF11 is specifically bound to the surface of the SPR sensor, directly changing the refractive index of the sensor surface and resulting in the shift of the SPR resonance valley. The rapid and accurate detection of GDF11 concentration was accomplished with the amount of movement of the SPR resonance valley. This sensor is expected to promote the development of GDF11 study in physiological and pathological processes [11,12].

## 2. Fiber Optic SPR Sensing Structure and Principle

### 2.1. Sensor Structure and Fabrication

The structure of the twisted single-mode fiber cladding SPR sensor was shown in Figure 1, which consists of a single-mode fiber on the left and a large core diameter step-index multimode fiber on the right. The single-mode optical fiber from left to right was composed of a light injection region and a torsion region made using hot-melt torsion micromachining technology and 2 cm long SPR sensing region. The 50-nm-thick gold film was coated on the surface of the SPR sensing region. The right end of the sensing region for the single-mode fiber was fused to the left end of the large core diameter step multimode fiber on the opposite axis. The broad-spectrum light was injected from the core of the left end of the single-mode fiber and transmitted to the twisted region. As the refractive index of the region was redistributed with the twisting of the fiber, the light in the core was coupled to the fiber cladding and excited higher-order modes of the fiber cladding. The light of cladding mode was transmitted to the right side, entered the sensing region cladding, contacted the 50 nm gold film on the surface of the sensing region cladding, and here the SPR phenomenon occurred. The light continued to transmit to the right side into a large core diameter step-index multimode fiber core. The SPR transmission spectrum was collected with the spectrometer. The optical fiber was divided into single-mode fiber and multimode fiber; multimode fiber was divided into graded-index multimode fiber and step-index multimode fiber. When constructing a twisted multimode fiber cladding SPR sensor, the single-mode fiber in the above structure was replaced by the graded index multimode fiber or step-index multimode fiber.

The sensor fabrication process was carried out according to the following steps. For the single-mode fiber twisted SPR sensing probe, we took a 1-m-long single-mode fiber (SMF-28e, Corning), removed the 5-cm-long coating layer from the middle part of the fiber using Miller pliers, and wiped it clean with a non-woven fabric moistened with alcohol to obtain the bare fiber. We used a fiber melt taper machine (full-featured research type-2, COUPLER) to prepare the twisted structure on the bare fiber. The twisted structure was prepared using a torch with an inner diameter of 10 mm, setting the height of the torch as 3 mm, the torsion speed as 12,000 μm/s, and the torsion time as 30 s. The twisted fiber with a pitch of 130 μm was prepared. The simulation and verification experiments investigated the effect of the pitch of the twisted region on the coupling of the optical field in the fiber core into the fiber cladding. When the pitch was too large, the optical field in the core was not sufficiently coupled to the cladding. When the pitch was too small, some of the light coupled to the fiber cladding leaked to the outside of the cladding, resulting in large energy loss. After optimization, a pitch parameter of 130 μm was chosen for a better coupling effect. The completed single-mode twisted fiber was removed from the fiber melting and taper machine, placed on the fiber cutting knife, and cut flat to 2 cm behind the twisted region.

The cutting single-mode twisted fiber was placed into the fiber fusion splicer (NT-200H, NOTEVIO) and fused coaxially with the large core diameter step-index multimode fiber (SI105/125-22/250, YOFC) after end face flattening handling. The fused probe was obtained from the above fiber splicer. A 300 μm inner diameter quartz sleeve wrapped around the light injection region, twisted region, and step-index multimode fiber region of the single-mode fiber, only exposing the 2-cm-long sensing region. The probe was clamped in a fiber optic rotary coating fixture of a plasma sputtering instrument (ETD-650MS, YLBT). 50 nm gold film was plated using a rotating probe to obtain the single-mode fiber twisted type SPR sensing probe. For a multimode fiber twisted type SPR sensing probe, the single mode fiber in the above process was replaced by a graded index multimode fiber (GI50/125-20/250, YOFC) or a step-index multimode fiber (SI50/125-22/250, YOFC).

### 2.2. The Simulation and Verification Experiments of Beam Transmission Path

In order to verify whether the twisted structure can effectively couple the light from various fiber cores into the fiber cladding, we used Rsoft simulation software to simulate the beam transmission path of various twisted fibers in this paper. The pitch parameter of the twisted region was set as 130 μm and the refractive index parameter of the cladding and core was set as 1.465 and 1.475, respectively.

The diameters of single-mode fiber cladding and core were set to 125 μm and 8 μm, respectively. The simulation results were shown in Figure 2a, which showed that the light in the single-mode core was effectively coupled into the fiber cladding at the torsion zone and was stably transmitted forward in the fiber cladding. The graded index multimode fiber cladding and core diameters were set to 125 μm and 50 μm, respectively, and the core refractive index type was set to “Diffused”. The simulation results were shown in Figure 2b: the light in the core was effectively coupled into the cladding; the cladding mode was stably transmitted forward. The step-index multimode fiber cladding and core diameters were set to 125 μm and 50 μm, respectively, and core refractive index type was set to “Step”. The simulation results were shown in Figure 2c: the light in the core was effectively coupled into the cladding and was stably transmitted forward in the fiber cladding.

To further investigate the effect of optical coupling of each type of twisted fiber to the cladding, we fabricated a single-mode twisted fiber with a pitch of 130 μm, graded an index multimode twisted fiber, and a step-index multimode twisted fiber. The microscopic experiments were conducted to compare the end face and side light fields of the three types of straight fibers and twisted fibers, respectively.

Figure 3a,b showed the microscopic physical photos, end face, and ambient light field photos of single-mode straight fiber and single-mode twisted fiber, respectively. As seen in Figure 3a, when injecting light into the single-mode straight fiber core, the light was always bound in the fiber core for transmission; no coupling from the core mode to the cladding mode occurs. As seen in Figure 3b, when injecting light into the single-mode twisted fiber core, the light was coupled between the core mode and the cladding mode and the fiber cladding mode was excited. The beam energy was still mainly distributed in the fiber core, which was consistent with the simulation results. Figure 3c,d showed the microscopic physical photos, end face, and side light field photos of the graded index multimode straight fiber and the graded index multimode twisted fiber, respectively. As seen in Figure 3c, the light energy in the fiber core was concentrated in the middle of the fiber core when injecting light into the core of the graded index multimode straight fiber, and the light was not coupled from the core mode to the cladding mode. As seen in Figure 3d, the light was coupled between the core mode and the cladding mode when injecting light into the core of the graded index multimode twisted fiber, and the fiber cladding mode was excited. However, the beam energy was still mainly distributed in the fiber core, which was consistent with the simulation results. Figure 3e,f showed the microscopic physical photos, end face, and side light field photos of the step multimode straight fiber and step-index multimode twisted fiber, respectively. As shown in Figure 3e, the light was transmitted in the fiber core when injecting light into the step-index multimode straight fiber core. When injecting light into the step-index multimode twisted fiber core, the energy of the light beam was distributed more uniformly into the fiber core and fiber cladding; the coupling efficiency from the fiber core to the fiber cladding was high, which agreed with the simulation results.

To further test the performance of the new cladding SPR sensor composed of three types of twisted fiber, three different types of twisted fibers were made into twisted SPR sensing probes according to the probe fabrication process and typical refractive index sensing tests of SPR sensors were performed.

## 3. Sensor Fiber Type Optimization and Surface Functionalization

### 3.1. Sensor Fiber Type Optimization

The twisted fiber SPR sensing probe was connected to a broad spectrum light source (HL2000, Ocean Optics) at the left end and a spectrometer (USB2000+, Ocean Optics) at the right end. The sensing area of the probe was placed on the experimental operation board and a standard refractive index solution was dropped over the sensing region of the sensing probe with a rubber-tipped dropper. The source light entered the fiber and was transmitted to the right. When it passed through the sensing region, the SPR phenomenon occurred. The light after SPR entered the thick-core multimode fiber on the right side was sent to the spectrometer to collect the SPR transmission spectrum and the computer processed the collected data. For three different types of twisted fiber SPR sensing probes, refractive index sensing experimental tests were performed. The sensing region of the probes was sequentially coated with standard refractive index solutions in the refractive index range of 1.333–1.385; the experimental test results are shown in Figure 4. Figure 4a–c showed the SPR transmission spectra results of the single-mode twisted fiber probe, graded index multimode twisted fiber probe, and the step-index multimode twisted fiber probe refractive index test, respectively. Figure 4d showed the relationship between the resonance wavelength and refractive index of the three sensing probes.

As seen in Figure 4, with the increase of refractive index of the test solution, the SPR resonance valleys of all three sensing probes moved to the long-wave direction, indicating that all three types of twisted fibers effectively constitute cladding SPR sensors for the sensing of the refractive index of the solution. As seen in Figure 4d, the average sensitivity of the single-mode twisted fiber probe was 2537.07 nm/RIU. The average sensitivity of the graded index multimode twisted fiber probe was only 2195.07 nm/RIU. The average sensitivity of the step-index multimode twisted fiber probe was the largest, reaching 3391.15 nm/RIU. Because of the excellent performance of the step-index multimode fiber twisted probe, we selected the twisted probe as the sensing probe for the GDF11 solution concentration detection experiment in this paper.

### 3.2. Surface Functionalization of the Sensing Probe

In order to obtain the high selection of the step-index multimode twisted fiber SPR sensing probe for GDF11 and better efficiency of specific binding of antigen–antibody on the sensor surface, it needed to be functionalized on the sensor surface; the functionalization process is shown in Figure 5a.

(1)The fiber optic twisted probe after plating gold film was placed in piranha solution (H_2_SO_4_:H_2_O_2_ = 3:1) to remove the dirt on the probe surface for 0.5 h. The probe was rinsed with deionized water and then blown dry. The fiber optic twisted probe was soaked for 3 h in the ZIF-67 solution with a concentration of 1 mg/mL. The ZIF-67 particles were well adsorbed on the gold film surface of the sensing probe. The scanning electron microscope photo was shown in Figure 5b;(2)The fiber optic twisted probe was loaded into a sealed reaction chamber. 3 mL of staphylococcal A protein (SPA) solution with a concentration of 1 μg/mL was injected into the reaction chamber with a syringe and stored at 10 °C for 3 h. The probe was rinsed with PBS buffer several times to remove the excess SPA residue on the surface and then air-dried naturally;(3)GDF11 antibody solution at a concentration of 50 μg/mL experienced carboxyl group activation by using EDC (0.2 mol/L)/NHS (0.05 mol/L). Activated GDF11 antibody solution was injected into the reaction chamber and stored at 10 °C for 3 h to ensure sufficient time for the antibody to bind to the sensor surface. The sensor was washed with PBS buffer to remove antibody molecules that were not immobilized on the sensor surface;(4)Then, 3 ml of bovine serum protein (BSA) at a concentration of 10 mg/mL was injected into the reaction chamber and stored at 10 °C for 0.5 h to occupy the non-specific binding sites on the sensor surface, followed by rinsing off the excess BSA using PBS buffer;(5)At this point, the surface functionalization of the step-index multimode twisted fiber SPR sensing probe was completed to obtain an SPR biosensor that can specifically detect GDF11. The sensor was further used for GDF11 concentration detection experiments.

## 4. Results

### 4.1. Experimental Test System Construction

The GDF11 concentration test device of the twisted fiber SPR sensing probe is shown in Figure 6. The broad-spectrum light source was injected into the step-index multimode fiber core of the probe from the left side and the sensing region was sealed in the reaction chamber after the functionalization of the central surface of the probe. The right side of the probe was connected to the spectrometer for the spectral acquisition of the light signal of the SPR effect and sent to the computer for data processing. The GDF11 solution to be measured was injected into the reaction chamber from the upper liquid inlet through the microfluidic syringe pump (LSP01-1A, Longer Pump); the measured solution flowed into the beaker from the lower liquid outlet hole.

### 4.2. Experimental Results of GDF11 Concentration Detection

The GDF11 antigen solution with a concentration of 50 ng/mL was diluted using PBS buffer to prepare GDF11 antigen solutions with concentrations of 10 ng/mL, 1 ng/mL, 100 pg/mL, 10 pg/mL, and 1 pg/mL, respectively. The experimental tests were performed using these five concentrations of GDF11 antigen solutions; the experimental results are shown in Figure 7. Figure 7a shows the tested SPR spectrum of GDF11 antigen solutions with different concentrations detected by step-index multimode twisted fiber SPR sensing probes. With the increase of the concentration of GDF11 antigen solution, the SPR resonance valley shifted towards the long-wave direction. In the detection range of lower concentrations, the movement of the SPR resonance valley was relatively large. It was due to the certain number of specific binding sites fixed on the sensor surface available. At low concentrations, the specific binding sites were sufficient. The refractive index of the sensor surface changes more, prompting the SPR resonance valley to move more. While with the increasing concentration of GDF11 antigen, the number of specific binding sites kept decreasing, leading to a smaller change of refractive index on the sensor surface and less movement of the SPR resonance valley.

Figure 7b showed the relation curve of SPR resonance wavelength and solution concentration. When the GDF11 concentration changed from 1 pg/mL to 10 ng/mL, the SPR resonance valley shifted 9.99 nm toward the long-wave direction. The detection sensitivity of the sensor for GDF11 antigen solution was 2.518 nm/lgC (lgC was the logarithm of GDF11 antigen solution concentration). According to the limit of detection (LOD) formula, LOD = λ/S (where λ is the resolution of the spectrometer and S is the sensitivity of the sensing probe). The LOD of the sensor was calculated to be only 0.34 pg/mL.

## 5. Discussion

The fiber is composed of a high refractive index fiber core in the center and a low refractive index cladding in circular cladding form. However, the SPR sensing structure cannot be realized due to the confinement of the cladding to the core field. In order to realize the light in the core to contact the gold film and construct the SPR sensing structure, it is often necessary to micro-process the fiber to remove the cladding or couple the light in the core to the cladding, such as corrosion, taper [13,14], laser etching [15], etc. The processing process is complex and the fiber probe becomes fragile. In addition, tilted FBG and LPG can also effectively couple the light in the fiber core to the fiber cladding to form a grating-structured cladding SPR sensor. These two types of grating SPR sensors are processed on the fiber core, which will not destroy the overall structure of the fiber and has the advantages of high physical strength and good repeatability. However, compared with conventional optical fiber SPR sensors, TFBG-SPR and LPG-SPR sensors have the disadvantages of low sensitivity and a complicated fabrication process [16,17]. In this paper, we used optical fiber hot-melt twisting technology to effectively couple the light from the core into the cladding. We constructed a new optical fiber cladding SPR sensor with fast and simple processing; the fiber structure was not damaged and still maintained strong physical strength [18,19]. In Table 1, we compared the sensing performances of different types of cladding sensors. It can be seen that the twisted SPR sensor proposed in this paper had higher physical strength and higher sensing sensitivity. The optical fibers as biosensing elements can enter narrow spaces such as blood vessels for detection work, such as protein concentration detection in a blood environment [20,21].

In addition, in this paper, different types of twisted fiber SPR sensing probes were optimized to obtain experimental probes (step-multimode twisted probe with a pitch of 130 μm) with higher refractive index sensing sensitivity. The results showed that cladding-type SPR sensors, based on step multimode twisted fibers, excited a higher order cladding mode. The sensors were more sensitive to external environmental refractive index changes and more easily identified the refractive index of the external environment of the gold film due to the specific binding of antibodies to the GDF11 antigen. The method realizes the accurate detection of GDF11 solution concentration.

In the surface functionalization of step-index multimode twisted fiber SPR sensing probes, the modification of the ZIF-67 layer outside the gold film can effectively increase the sensitivity of the fiber sensor. This is due to its unique properties: under the excitation of transmitted light in the fiber, the electrons in ZIF-67 are continuously transferred to the gold film, resulting in enhanced electric field coupling strength on the surface of the gold film, thus increasing the sensitivity of the sensor. The subsequent modified SPA layer, which has the property of binding to the crystallizable segment (Fc) of the GDF11 antibody, can realize the function of immobilizing the GDF11 antibody and providing sites for specific binding of GDF11 antigen. After occupying the non-specific binding sites on the sensor surface by using BSA solution, it prevents SPR resonance valley movement with the change of the refractive index of the sensor surface due to the non-specific binding sites in the experiment. This is one of the reasons why the sensor can achieve specific detection of the GDF11 antigen–antibody. In order to show the performance of the sensor more intuitively, the performance of different biochemical sensors is compared in Table 2. It can be seen that the twisted SPR sensor proposed in this paper had a large detection range and a low detection limit, indicating that the sensor has the potential to continue to promote the research of GDF11.

## 6. Conclusions

In this paper, a twisted fiber based on a cladding-type SPR sensor is proposed to achieve high-accuracy detection of GDF11 concentration. The twisted SPR sensors on different types of optical fibers are constructed by using a hot-melt torsion technique. After the optimization of the fiber type, the average sensitivity of the step-index multimode twisted SPR sensor for refractive index detection reaches 3391.15 nm/RIU. After the functionalization of the GDF11 antibody on the sensor surface, the specific detection of the GDF11 antigen is achieved with a detection sensitivity of 2.518 nm/lgC and a detection limit of 0.34 pg/mL. This sensor has potential to play an important role in the study of the GDF11 mechanism. It is expected to enable specific concentration detection of other members of the growth differentiation factor family using a targeted modification of the probe surface with different antibodies. The diameter of the probe is only 125 μm, which realizes the detection of microscopic solutions, as well as access to small spaces such as blood vessels for online detection work.

## Figures and Tables

**Figure 1 micromachines-13-01914-f001:**
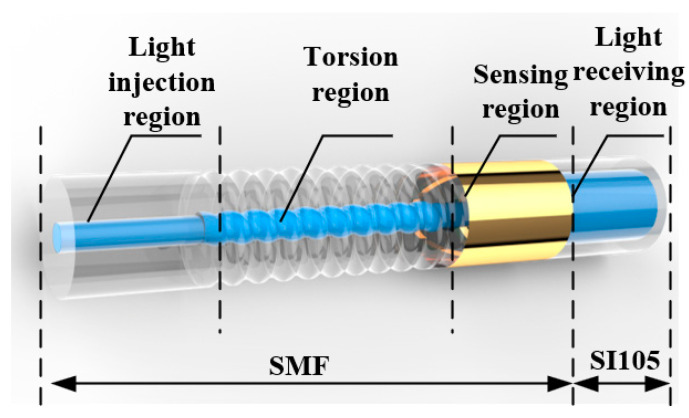
Structure of twisted fiber cladding SPR sensor.

**Figure 2 micromachines-13-01914-f002:**
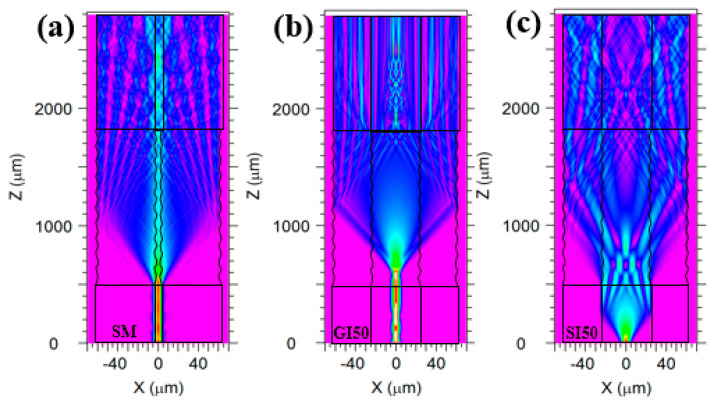
The simulation results of beam transmission path in different types of twisted fiber: (**a**) single−mode twisted fiber; (**b**) graded index multimode twisted fiber; (**c**) step−index multimode twisted fiber.

**Figure 3 micromachines-13-01914-f003:**
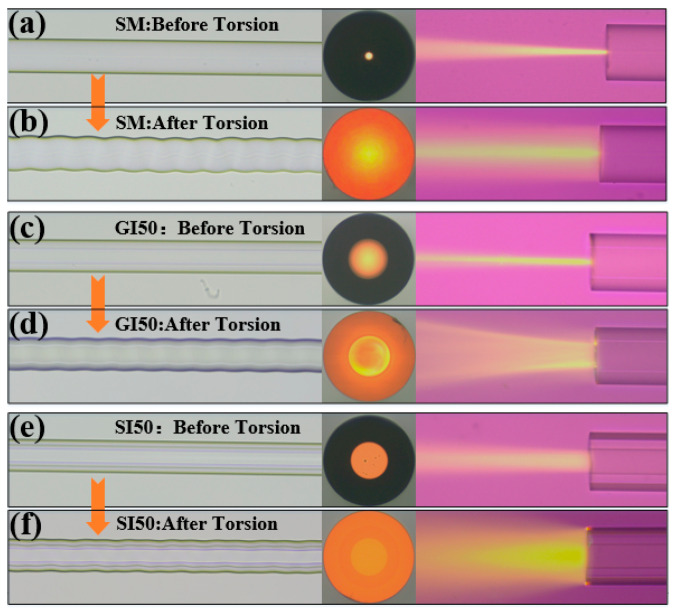
Microscopic physical photos, end face, and side emitted light field photos of different types of straight and twisted optical fibers: (**a**) single-mode straight fiber; (**b**) single-mode twisted fiber; (**c**) graded index multimode straight fiber; (**d**) graded index multimode twisted fiber; (**e**) step-index multimode straight fiber; (**f**) step-index multimode twisted fiber.

**Figure 4 micromachines-13-01914-f004:**
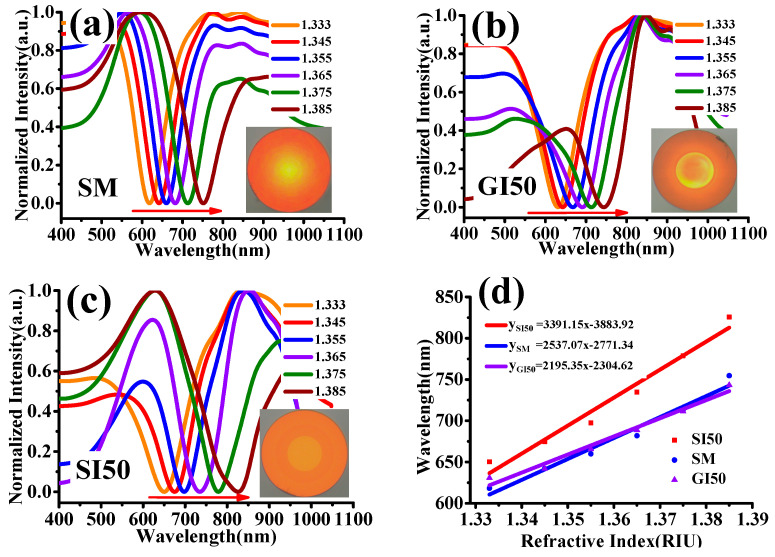
Refractive index sensing test results of different types of twisted fiber SPR probes: (**a**) single-mode twisted probe; (**b**) graded index multimode twisted probe; (**c**) step-index multimode twisted probe; (**d**) resonant wavelength versus refractive index curve.

**Figure 5 micromachines-13-01914-f005:**
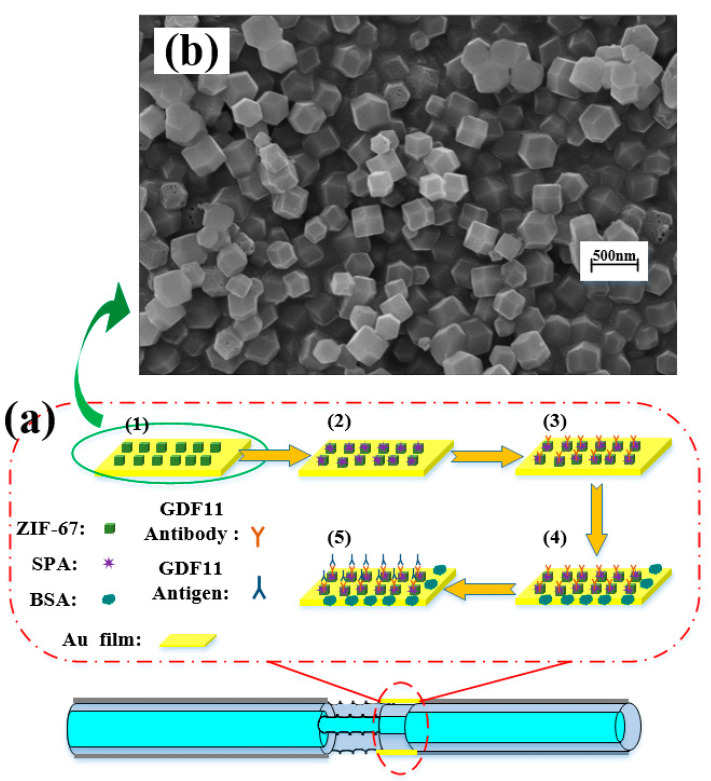
Schematic diagram of the surface functionalization of the sensing probe: (**a**) surface modification process of the fiber SPR biosensor; (**b**) scanning electron microscope photo after ZIF-67 modification.

**Figure 6 micromachines-13-01914-f006:**
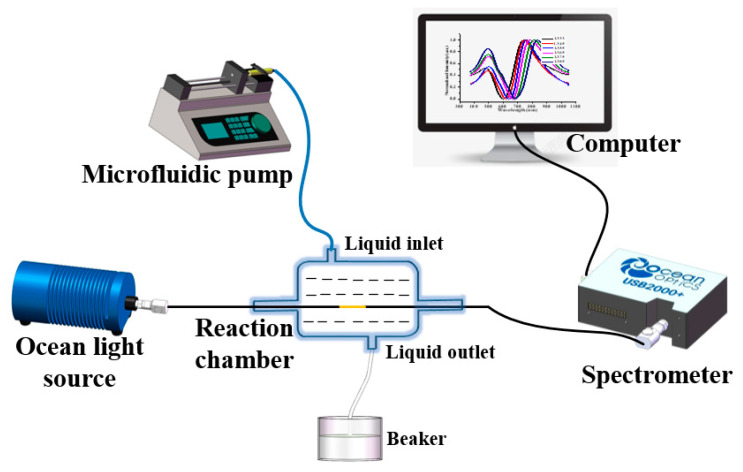
GDF11 concentration testing device of twisted fiber SPR sensing probe.

**Figure 7 micromachines-13-01914-f007:**
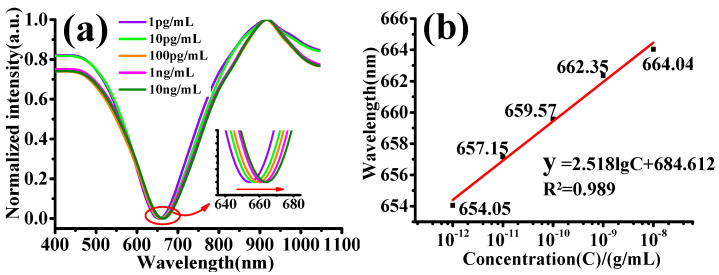
The test results of GDF11 concentration detected by twisted fiber SPR sensing probe: (**a**) SPR spectra of different concentrations of GDF11 antigen solution; (**b**) The relation curve of SPR resonance wavelength and solution concentration.

**Table 1 micromachines-13-01914-t001:** Performance comparison of different cladding SPR sensors.

Sensing Principle	PROCESSING METHOD	Refractive Index Sensing Performance	Physical Strength	Reference
SPR	Taper	1780 nm/RIU	Weak	[14]
SPR	Laser beam modulation engraving	2896.4 nm/RIU	Weak	[15]
TFBG-SPR	Phase mask	1023 nm/RIU	Strong	[22]
LPG-SPR	Laser beam modulation engraving	1600 nm/RIU	Strong	[23]
SPR	Hot-melt torsion	3391.15 nm/RIU	Strong	This work

**Table 2 micromachines-13-01914-t002:** Performance comparison of different optical fiber biochemical sensors.

Sensing Principle	Detection Substances	Detection Range	Detection Limitation	Reference
SPR	IgG	2 mg/mL–100 mg/mL	0.90 μg/mL	[24]
SPR	MMP-9	10 ng/mL–200 ng/mL	8 pg/mL	[25]
SPR	DNA	10 pM–100 pM	10 pM	[26]
TFBG-SPR	Hg^2+^	10 pM–1 mM	3.073 pM	[16]
LPG-SPR	As^3+^	0–0.2 ppb	0.04 ppb	[17]
LSPR	DNA	100 pM–1 μM	67 pM	[27]
SPR	GDF11	1 pg/mL–10 ng/mL	0.34 pg/mL	This work

## Data Availability

Not applicable.

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
