# Peer review of "Twisted Fiber Optic SPR Sensor for GDF11 Concentration Detection"

_micromachines, 2022, doi:10.3390/mi13111914_

Round 1

Reviewer 1 Report

The authors presented an optical fibre based sensor for detection and measurment of concentration of GDF11. Devices and medical technology as presented no only simplifies clinical diagnostics but also enhances point-of-care capabilities of small health systems. 

I have 2 general comments and 2 specifics. 

General:

1.     I suggest to the authors that they should have a native English speaker copywrite the article or have professional writing service edit the article. There are multiple points of the paper that i must read and re-read in order to comprehend the semantics of the text correctly. 

2.     Authors discussed the benefits of twisted fibres over cladding removed or modified fibres to couple core modes into cladding modes. However, there are another major branch of cladding coupling technic using gratings such as tilted FBGs or LPGs that should not be over looked and can be more repeatable.

Specifics: 

1.     The authors seem to have conducted the experiments in different concentration of analytes in succession. This would indicate that bonding sites that were previously taken up in the previous sample is still bound to the sensor. Would this not affect the accuracy of subsequent measurements. Are there not technics to refresh the sensor after each measurement?

2.     If x axis of figure 7b correctly labelled? Is the concentration the absolute concentration or LogC?

3.     It might be due to language. I was still unsure at discussion which sensor configuration was tested with and what are the selection criteria.

Overall, I think this is very useful work. Thank you for contributing to bio-photonics

Author Response

Reviewer 1:

The authors presented an optical fibre based sensor for detection and measurment of concentration of GDF11. Devices and medical technology as presented no only simplifies clinical diagnostics but also enhances point-of-care capabilities of small health systems.I have 2 general comments and 2 specifics.

1.I suggest to the authors that they should have a native English speaker copywrite the article or have professional writing service edit the article. There are multiple points of the paper that i must read and re-read in order to comprehend the semantics of the text correctly.

Answer: Thanks for the suggestion. We have revised the language errors in the article according to the suggestion.

The details are as follows :

The sentence “a novel fiber optic cladding SPR specific detection sensor was constructed to achieve rapid detection of GDF11 concentration.”has changed into “a novel fiber optical cladding SPR sensor was constructed to achieve rapid detection of GDF11 concentration.

The sentence“The light continued to transmitting to the right side into coarse-core step index multimode fiber core.”has changed into“The light continued to transmitting to the right side into large core diameter step index multimode fiber core.

The sentence“We used a fiber melt taper machine (full-featured research type-2, COUPLER) to prepare the torsional structure on the bare fiber”has changed into“We used a fiber melt taper machine (full-featured research type-2, COUPLER) to prepare the twisted structure on the bare fiber.”

  1. Authors discussed the benefits of twisted fibers over cladding removed or modified fibers to couple core modes into cladding modes. However, there are another major branch of cladding coupling technic using gratings such as tilted FBGs or LPGs that should not be over looked and can be more repeatable.

Answer: Thanks for the suggestion. Tilted FBG and LPG can also effectively couple the light in the fiber core to the fiber cladding to form a grating-structured cladding SPR sensor. These two types of grating SPR sensors are fabricated on the fiber core without damaging the overall structure of the fiber. These two types of grating SPR sensors have the advantages of high physical strength and good repeatability. However, compared with conventional optical fiber SPR sensor, TFBG-SPR and LPG-SPR sensors have the disadvantages of low sensitivity and complicated fabrication process. According to your suggestion, we have added this section to the discussion part.

We have added the above content to the discussion part, as follows (Line 308 to 321):

The fiber is composed of a high refractive index fiber core in the center and a low refractive index cladding in circular cladding form. However, the SPR sensing structure cannot be realized due to the confinement of the cladding to the core field. In order to realize the light in the core to contact the gold film and construct the SPR sensing structure, it is often necessary to micro-process the fiber to remove the cladding or couple the light in the core to the cladding, such as corrosion, taper [13,14], laser etching [15], etc. The processing process is complex and the fiber probe becomes fragile. In addition, tilted FBG and LPG can also effectively couple the light in the fiber core to the fiber cladding to form a grating-structured cladding SPR sensor. These two types of grating SPR sensors are processed on the fiber core, which will not destroy the overall structure of the fiber, and have the advantages of high physical strength and good repeatability. However, compared with conventional optical fiber SPR sensor, TFBG-SPR and LPG-SPR sensors have the disadvantages of low sensitivity and complicated fabrication process [16,17].

  1. The authors seem to have conducted the experiments in different concentration of analytes in succession. This would indicate that bonding sites that were previously taken up in the previous sample is still bound to the sensor. Would this not affect the accuracy of subsequent measurements. Are there not technics to refresh the sensor after each measurement?

Answer: Thanks for the suggestion. This does not affect the accuracy of subsequent measurements because the concentration of the GDF11 antigen solution to be tested in the experiment is very low, only 1pg/mL-10ng/mL. When the surface of the sensor is functionalized, the concentration of GDF11 antibody used is 50 μg/mL, which provides enough antibody molecules to specifically bind to GDF11 antigen molecules in the solution to be tested, so each detection can ensure the corresponding accuracy. After each measurement, there is no technology to refresh the sensor. The combination between antigen and antibody has been combined and the use of strong base can destroy the combination between GDF11 antigen and antibody. The measurement is carried out from low concentration to high concentration. After each measurement, only PBS buffer washing is carried out to remove the residual antigen molecules that are not bound on the surface of the sensor. The method can avoid these residual antigen molecules affecting the concentration of the subsequent solution to be measured, resulting in experimental error.

  1. If x axis of figure 7b correctly labelled? Is the concentration the absolute concentration or LogC?

Answer: Thanks for the suggestion. The x-axis in Figure 7b is correct and the concentration is absolute value. Because the preparation of solution concentration range is too large, we change the abscissa to logarithmic coordinates to facilitate the display of results, in the process of data processing. The abscissa unit has no effect and is still the absolute concentration.

5.It might be due to language. I was still unsure at discussion which sensor configuration was tested with and what are the selection criteria.

Answer: Thanks for the suggestion. In this paper, the SI50 twisted fiber probe with a pitch of 130μm is selected for experimental testing. The reason for choosing this probe is that the probe has the best refractive index sensitivity in the comparison of refractive index sensing performance of single-mode twisted fiber probe, GI50 twisted fiber probe and SI50 twisted fiber probe. It means that the probe can more sensitively sense the change of refractive index in the external environment and is more suitable for the specific detection of GDF11. The pitch parameter of 130μm is selected because the optical field coupling characteristics of the twisted fiber are the best under this parameter.

The parameters of the sensing probe for the GDF11 detection experiment are discussed. The specific contents of this part are as follows (Line 332 to 334):

In addition, in this paper, different types of twisted fiber SPR sensing probes were optimized to obtain experimental probes (step-multimode twisted probe with pitch of 130 μm) with higher refractive index sensing sensitivity. The results showed cladding-type SPR sensors, based on step-multimode twisted fibers, excited a higher order cladding mode.

Article

Twisted fiber optic SPR sensor for GDF11 concentration detection

Yong Wei 1, Ze Ran 1, Rui Wang 1, Zhuo Ren 1, Chun-Lan Liu 1, Chun-Biao Liu 1, Chen Shi 1, Chen Wang 2 and Yong-Hui Zhang 2,*

Citation: Lastname, F.; Lastname, F.; Lastname, F. Title. Micromachines 2022, 13, x. https://doi.org/10.3390/xxxxx

Academic Editor: Firstname Lastname

Received: date

Accepted: date

Published: date

Publisher’s Note: MDPI stays neutral with regard to jurisdictional claims in published maps and institutional affiliations.

Copyright: © 2022 by the authors. Submitted for possible open access publication under the terms and conditions of the Creative Commons Attribution (CC BY) license (https://creativecommons.org/licenses/by/4.0/).

1 College of Electronic&Information Engineering, Chongqing Three Gorges University, Chongqing 404100, China

2 Chongqing Key Laboratory of Development and Utilization of Genuine Medicinal Materials in Three Gorges Reservoir Area, Chongqing Three Gorges Medical College, Chongqing 404120, China

* Correspondence: [email protected]

Abstract: There are few methods and insufficient accuracy for growth differentiation factor 11 (GDF11) concentration detection. In this paper, we designed a twisted fiber cladding surface plasmon resonance (SPR) sensor, which can achieve a high precision detection of GDF11 concentration. The new structure of the fiber cladding SPR sensor was realized by coupling the light in the fiber core to the cladding through fiber thermal fusion twisting micromachining technology, and a series of functionalized modifications were made to the sensor surface to obtain a fiber sensor capable of GDF11 specific recognition. The experimental results showed when GDF11 antigen concentration was 1 pg/ml-10 ng/ml, the sensor had a detection sensitivity of 2.518 nm/lgC, the detection limit of 0.34 pg/ml, and had a good log-linear relationship. The sensor is expected to play a role in the rapid and accurate concentration detection of pathological study for growth differentiation factors.

Keywords: GDF11 concentration detection; twisted fiber optic SPR sensor; SPR biosensor, growth differentiation factor specific detection

  1. Introduction

Growth differentiation factor 11 (GDF11), as a member of transforming growth factor β (TGFβ) protein, is involved in many physiological and pathological processes. However, the studies on GDF11 function by different teams have obtained contradictory results. On the one hand, GDF11 is reported to involve in the regulation of the aging process in the body and plays a protective role in the function of brain, skeletal muscle and heart [1,2]. Researchers at the Harvard Stem Cell Institute found that GDF11 levels decreased with age and that GDF11 reversed aging-related cardiovascular remodeling, muscle aging, and neural senescence [3]. However, studies from other laboratories have reached the opposite conclusion that GDF11 levels increased with age and GDF11 inhibited skeletal muscle regeneration [4]. At the same time, contradictory findings have been seen in studies of GDF11 on tumors [5]. One of the important reasons is the lack of uniformity in the detection standards and the lack of precision of the assay of GDF11 concentration by each team. Therefore, accurate and rapid detection method of GDF11 concentration is important in the study of GDF11 in physiological and pathological processes.

In the past studies, less work has been reported on GDF11 concentration detection method. There are two main methods to perform GDF11 concentration assays. The first method is the enzyme-linked immunosorbent assay (ELISA). The antibody is labeled by the combination of the enzyme molecule and the antibody molecule. The labeled antibody is specifically combined with the antigen. After the substrate solution is dropped, the substrate will undergo a color reaction under the action of the enzyme. The color reaction can be analyzed by the ELISA detector [6]. However, this assay has strict requirements for the activity of the enzyme molecules to be labeled, which has an important impact on the accuracy of the detection of the concentration of the analyte to be measured. In addition, the interpretation of the chromogenic reaction by ELISA assay is limited by the accuracy of the instrument, which can also have an impact on the determination of GDF11 concentration. The second method is fluorescence immunoassay. The antibody (or antigen) is labeled with fluorescent materials, and after specific binding with the antigen (or antibody), fluorescence photos are taken by fluorescence microscope. The properties, localization and concentration of the antigen or antibody are analyzed by setting appropriate measurement indicators [7]. However, the strength of the fluorescence signal, poor labeling and high background noise can directly affect the interpretation of the analysis results and inaccurate calibration of the concentration of the analyte to be measured. Therefore, it becomes an urgent problem to develop a new detection method for the direct, accurate and real-time detection of GDF11 concentrations.

Optical fiber surface plasmon resonance (SPR) sensing technology, which has the advantages of non-labeling test sample, real time online detection, and high sensing sensitivity, is widely used in biochemical sensing [8-10]. The detection principle is that the specific binding of antigen and antibody molecules on the sensor surface to change the refractive index of the external environment of the gold film on the sensor surface. The SPR phenomenon is very sensitive to the changes in the refractive index of the environment, which causes the SPR resonance valley to shift. Therefore, the movement of the SPR resonance valley can directly and sensitively surface features the change of the test sample concentration.

In this paper, a novel fiber optic cladding SPR sensor was constructed to achieve rapid detection of GDF11 concentration. The key problem for cladding SPR sensor was effectively coupling the light in fiber core to the cladding. It was solved by using the fiber hot-melt torsion micromachining technology. The cladding surface was plated with 50nm gold film, which constitutes the cladding SPR sensing structure that is easy to be used for biochemical detection. The antigen-antibody of GDF11 specifically bound to the surface of SPR sensor, directly changing the refractive index of the sensor surface and resulting in the shift of SPR resonance valley. The rapid and accurate detection of GDF11 concentration was accomplished by the amount of movement of the SPR resonance valley. This sensor is expected to promote the development of GDF11 study in physiological and pathological processes [11,12].

  1. Fiber optic SPR sensing structure and principle

2.1. Sensor structure and fabrication

The structure of the twisted single-mode fiber cladding SPR sensor was shown in Figure 1, which consists of a single-mode fiber on the left and a large core diameter step index multimode fiber on the right. Single-mode optical fiber from left to right was composed of light injection region, torsion region made by hot melt torsion micromachining technology and 2 cm long SPR sensing region. 50-nm-thick gold film was coated on the surface of the SPR sensing region. The right end of the sensing region for the single-mode fiber was fused to the left end of the large core diameter step multimode fiber on the opposite axis. The broad-spectrum light was injected from the core of the left end of the single-mode fiber and transmitted to the twisted region. As the refractive index of the region was redistributed with the twisting of the fiber, the light in the core was coupled to the fiber cladding and excites the fiber cladding higher-order mode. The light of cladding mode transmitted to the right side, entered the sensing region cladding, contacted the 50 nm gold film on the surface of the sensing region cladding and the SPR phenomenon occurred. The light continued to transmitting to the right side into large core diameter step index multimode fiber core. The SPR transmission spectrum was collected by the spectrometer. Optical fiber was divided into single-mode fiber and multimode fiber, and multimode fiber was divided into graded index multimode fiber and step index multimode fiber. When constructing a twisted multimode fiber cladding SPR sensor, the single-mode fiber in the above structure was replaced by the graded index multimode fiber or step index multimode fiber.

Figure 1. Structure of twisted fiber cladding SPR sensor.

The sensor fabrication process was carried out according to the following steps. For single-mode fiber twisted SPR sensing probe, we took a 1m-long single-mode fiber (SMF-28e, Corning), removed the 5 cm-long coating layer from the middle part of the fiber using Miller pliers, wiped it clean with a non-woven fabric moistened with alcohol to obtain the bare fiber. We used a fiber melt taper machine (full-featured research type-2, COUPLER) to prepare the twisted structure on the bare fiber. The twisted structure was prepared by using a torch with an inner diameter of 10 mm, setting the height of the torch as 3 mm, the torsion speed as 12000 μm/s and the torsion time as 30 s. The twisted fiber with a pitch of 130 μm was prepared (The simulation and verification experiments investigated the effect of the pitch of the twisted region on the coupling of the optical field in the fiber core into the fiber cladding. When the pitch was too large, the optical field in the core was not sufficiently coupled to the cladding. When the pitch was too small, some of the light coupled to the fiber cladding leaked to the outside of the cladding, resulting in large energy loss. After optimization, a pitch parameter of 130 μm was chosen for the better coupling effect). The completed single-mode twisted fiber was removed from the fiber melting and taper machine, placed on the fiber cutting knife, and cut flat to 2 cm behind the twisted region.

The cutting single-mode twisted fiber was placed into the fiber fusion splicer (NT-200H, NOTEVIO) and fused coaxially with the large core diameter step index multimode fiber (SI105/125-22/250, YOFC) after end face flattening handling. The fused probe was obtained from the above fiber splicer. A 300 μm inner diameter quartz sleeve wrapped the light injection region, twisted region and step index multimode fiber region of the single-mode fiber, only exposing the 2cm long sensing region. The probe was clamped in a fiber optic rotary coating fixture of a plasma sputtering instrument (ETD-650MS, YLBT). 50nm gold film was plated by rotate probe to obtain the single-mode fiber twisted type SPR sensing probe. For a multimode fiber twisted type SPR sensing probe, the single mode fiber in the above process was replaced by a graded index multimode fiber (GI50/125-20/250, YOFC) or a step index multimode fiber (SI50/125-22/250, YOFC).

2.2. The simulation and verification experiments of beam transmission path

In order to verify whether the twisted structure can effectively couple the light from various fiber cores into the fiber cladding, we used Rsoft simulation software to simulate the beam transmission path of various twisted fibers in this paper. The pitch parameter of the twisted region was set as 130 μm and the refractive index parameter of the cladding and core was set as 1.465 and 1.475, respectively.

The diameters of single-mode fiber cladding and core were set to 125 μm and 8 μm, respectively. The simulation results were shown in Figure 2(a), which showed that the light in the single-mode core was effectively coupled into the fiber cladding at the torsion zone and was stably transmitted forward in the fiber cladding. The graded index multimode fiber cladding and core diameters were set as 125 μm and 50 μm, respectively, and the core refractive index type was set as “Diffused”. The simulation results were shown in Figure 2(b): the light in the core was effectively coupled into the cladding, and the cladding mode was stably transmitted forward. Step index multimode fiber cladding and core diameters were set to 125 μm and 50 μm, respectively, core refractive index type was set as “Step”. The simulation results were shown in Figure 2(c): the light in the core was effectively coupled into the cladding and was stably transmitted forward in the fiber cladding.

Figure 2. The simulation results of beam transmission path in different types of twisted fiber. (a) single-mode twisted fiber (b) graded index multimode twisted fiber (c) step index multimode twisted fiber.

To further investigate the effect of optical coupling of each type of twisted fiber to the cladding, we fabricated single-mode twisted fiber with a pitch of 130 μm, graded index multimode twisted fiber and step index multimode twisted fiber. The microscopic experiments were conducted to compare the end face and side light fields of the three types of straight fibers and twisted fibers, respectively.

Figure 3(a) and (b) showed the microscopic physical photos, end face and ambient light field photos of single-mode straight fiber and single-mode twisted fiber, respectively. As Figure 3(a), when injecting light into the single-mode straight fiber core, the light was always bound in the fiber core for transmission, and no coupling from the core mode to the cladding mode occurs. As Figure 3(b), when injecting light into the single-mode twisted fiber core, the light was coupled between the core mode and the cladding mode and the fiber cladding mode was excited. The beam energy was still mainly distributed in the fiber core, which was consistent with the simulation results. Figure 3(c) and (d) showed the microscopic physical photos, end face and side light field photos of the graded index multimode straight fiber and the graded index multimode twisted fiber, respectively. As Figure 3(c), the light energy in the fiber core was concentrated in the middle of the fiber core when injecting light into the core of the graded index multimode straight fiber, and the light was not coupled from the core mode to the cladding mode. As Figure 3(d), the light was coupled between the core mode and the cladding mode, when injecting light into the core of the graded index multimode twisted fiber, and the fiber cladding mode was excited. But the beam energy was still mainly distributed in the fiber core, which was consistent with the simulation results. Figure 3(e) and (f) showed the microscopic physical photos, end face and side light field photos of the step multimode straight fiber and step index multimode twisted fiber, respectively. As shown in Figure 3(e), the light was transmitted in the fiber core when injecting light into the step index multimode straight fiber core. When injecting light into the step index multimode twisted fiber core, the energy of the light beam was distributed more uniformly into the fiber core and fiber cladding, and the coupling efficiency from the fiber core to the fiber cladding was high, which agreed with the simulation results.

Figure 3. Microscopic physical photos, end face and side emitted light field photos of different types of straight and twisted optical fibers. (a) single-mode straight fiber (b) single-mode twisted fiber (c) graded index multimode straight fiber (d) graded index multimode twisted fiber (e) step index multimode straight fiber (f) step index multimode twisted fiber.

To further test the performance of the new cladding SPR sensor composed of three types of twisted fiber, three different types of twisted fibers were made into twisted SPR sensing probes according to the probe fabrication process, and typical refractive index sensing tests of SPR sensors were performed.

  1. Sensor fiber type optimization and surface functionalization

3.1. Sensor fiber type optimization

The twisted fiber SPR sensing probe was connected to a broad spectrum light source (HL2000, Ocean Optics) at the left end and a spectrometer (USB2000+, Ocean Optics) at the right end. The sensing area of the probe was placed on the experimental operation board, and a standard refractive index solution can be dropped over the sensing region of the sensing probe with a rubber-tipped dropper. The source light entered the fiber and transmitted along the right direction. When it passed through the sensing region, the SPR phenomenon occurred. The light after SPR entered the thick-core multimode fiber on the right side and was sent to the spectrometer to collect the SPR transmission spectrum, and the computer processed the collected data. For three different types of twisted fiber SPR sensing probes, refractive index sensing experimental tests were performed. The sensing region of the probes were sequentially coated with standard refractive index solutions in the refractive index range of 1.333-1.385, and the experimental test results were shown in Figure 4. Figure 4(a), (b) and (c) showed the SPR transmission spectra results of single-mode twisted fiber probe, graded index multimode twisted fiber probe and step index multimode twisted fiber probe refractive index test, respectively. Figure 4(d) showed the relationship between the resonance wavelength and refractive index of the three sensing probes.

Figure 4. Refractive index sensing test results of different types of twisted fiber SPR probes. (a) single-mode twisted probe (b) graded index multimode twisted probe (c) step index multimode twisted probe d) resonant wavelength versus refractive index curve.

As Figure 4, with the increase of refractive index of the test solution, the SPR resonance valleys of all three sensing probes moved to the long-wave direction, indicating that all three types of twisted fibers effectively constitute cladding SPR sensors for the sensing of the refractive index of the solution. As Figure 4(d), the average sensitivity of the single-mode twisted fiber probe was 2537.07 nm/RIU. The average sensitivity of the graded index multimode twisted fiber probe was only 2195.07 nm/RIU. The average sensitivity of the step index multimode twisted fiber probe was the largest, reaching 3391.15 nm/RIU. Because of the excellent performance of the step index multimode fiber twisted probe, we selected the twisted probe as the sensing probe for the GDF11 solution concentration detection experiment in this paper.

3.2. Surface functionalization of the sensing probe

In order to obtain the highly selection of the step index multimode twisted fiber SPR sensing probe for GDF11 and better efficiency of specific binding of antigen antibody on the sensor surface, it needed to be functionalized on the sensor surface, and the functionalization process was shown in Figure 5(a).

(1) The fiber optic twisted probe after plating gold film was placed in piranha solution (H2SO4:H2O2=3:1) to remove the dirt on the probe surface for 0.5 h. The probe was rinsed with deionized water and then blown dry. The fiber optic twisted probe was soaked for 3 h in the ZIF-67 solution with a concentration of 1mg/ml. The ZIF-67 particles were well adsorbed on the gold film surface of sensing probe. The scanning electron microscope photo was shown in Figure 5(b).

(2) The fiber optic twisted probe was loaded into a sealed reaction chamber. 3 ml of staphylococcal A protein (SPA) solution with a concentration of 1 μg/ml was injected into the reaction chamber with a syringe and stored at 10°C for 3 h. The probe was rinsed with PBS buffer several times to remove the excess SPA residue on the surface and then air-dried naturally.

(3) GDF11 antibody solution at a concentration of 50 μg/ml experienced carboxyl group activation by using EDC (0.2 mol/l)/NHS (0.05 mol/l). Activated GDF11 antibody solution was injected into the reaction chamber and stored at 10°C for 3 h to ensure sufficient time for the antibody to bind to the sensor surface. The sensor was washed with PBS buffer to remove antibody molecules that were not immobilized on the sensor surface.

(4) 3ml of bovine serum protein (BSA) at a concentration of 10 mg/ml was injected into the reaction chamber and stored at 10°C for 0.5 h to occupy the non-specific binding sites on the sensor surface, followed by rinsing off the excess BSA using PBS buffer.

(5) At this point, the surface functionalization of the step index multimode twisted fiber SPR sensing probe was completed to obtain an SPR biosensor that can specifically detect GDF11. The sensor was further used for GDF11 concentration detection experiments.

Figure 5. Schematic diagram of the surface functionalization of the sensing probe. (a) surface modification process of the fiber SPR biosensor (b) scanning electron microscope photo after ZIF-67 modification.

  1. Results

4.1. Experimental test system construction

The GDF11 concentration test device of the twisted fiber SPR sensing probe was shown in Figure 6. The broad-spectrum light source was injected into the step index multimode fiber core of the probe from the left side, and the sensing region was sealed in the reaction chamber after the functionalization of the central surface of the probe. The right side of the probe was connected to the spectrometer for spectral acquisition of the light signal of the SPR effect and sent to the computer for data processing. The GDF11 solution to be measured was injected into the reaction chamber from the upper liquid inlet through the microfluidic syringe pump (LSP01-1A, Longer Pump), and the measured solution flowed into the beaker from the lower liquid outlet hole.

Figure 6. GDF11 concentration testing device of twisted fiber SPR sensing probe.

4.2. Experimental results of GDF11 concentration detection

The GDF11 antigen solution with a concentration of 50 ng/ml was diluted using PBS buffer to prepare GDF11 antigen solutions with concentrations of 10 ng/ml, 1 ng/ml, 100 pg/ml, 10 pg/ml, and 1 pg/ml, respectively. The experimental tests were performed using these five concentrations of GDF11 antigen solutions, and the experimental results were shown in Figure 7. Figure 7(a) showed the tested SPR spectrum of GDF11 antigen solutions with different concentrations detected by step index multimode twisted fiber SPR sensing probes. With the increase of the concentration of GDF11 antigen solution, the SPR resonance valley shifted towards the long-wave direction. In the detection range of lower concentrations, the movement of SPR resonance valley was relatively large. It was due to the certain number of specific binding sites fixed on the sensor surface available. At low concentrations, the specific binding sites were sufficient. The refractive index of the sensor surface changes more, prompting the SPR resonance valley to move more. While with the increasing concentration of GDF11 antigen, the number of specific binding sites kept decreasing, leading to the smaller change of refractive index on the sensor surface and the less movement of the SPR resonance valley.

Figure 7(b) showed the relation curve of SPR resonance wavelength and solution concentration. When the GDF11 concentration changed from 1 pg/ml to 10 ng/ml, the SPR resonance valley shifted 9.99 nm toward the long-wave direction. The detection sensitivity of the sensor for GDF11 antigen solution was 2.518 nm/lgC (lgC was the logarithm of GDF11 antigen solution concentration). According to the limit of detection (LOD) formula, LOD = λ/S (where λ is the resolution of the spectrometer and S is the sensitivity of the sensing probe). The LOD of the sensor was calculated to be only 0.34 pg/ml

Figure 7. The test results of GDF11 concentration detected by twisted fiber SPR sensing probe. (a) SPR spectra of different concentrations of GDF11 antigen solution (b) The relation curve of SPR resonance wavelength and solution concentration.

  1. Discussion

The fiber is composed of a high refractive index fiber core in the center and a low refractive index cladding in circular cladding form. However, the SPR sensing structure cannot be realized due to the confinement of the cladding to the core field. In order to realize the light in the core to contact the gold film and construct the SPR sensing structure, it is often necessary to micro-process the fiber to remove the cladding or couple the light in the core to the cladding, such as corrosion, taper [13,14], laser etching [15], etc. The processing process is complex and the fiber probe becomes fragile. In addition, tilted FBG and LPG can also effectively couple the light in the fiber core to the fiber cladding to form a grating-structured cladding SPR sensor. These two types of grating SPR sensors are processed on the fiber core, which will not destroy the overall structure of the fiber, and have the advantages of high physical strength and good repeatability. However, compared with conventional optical fiber SPR sensor, TFBG-SPR and LPG-SPR sensors have the disadvantages of low sensitivity and complicated fabrication process [16,17]. In this paper, we used optical fiber hot melt twisting technology to effectively couple the light from the core into the cladding. We constructed a new optical fiber cladding SPR sensor with fast and simple processing, and the fiber structure was not damaged and still maintained strong physical strength [18,19]. In table 1, we compared the sensing performances of different types of cladding sensors. It can be seen that the twisted SPR sensor proposed in this paper had higher physical strength and higher sensing sensitivity. The optical fibers as biosensing elements can enter into narrow spaces such as blood vessels for detection work, such as protein concentration detection in blood environment [20,21].

Table-1. Performance comparison of different cladding SPR sensors.

Sensing principle

Processing method

Refractive index sensing performance

Physical strength

Reference

SPR

Taper

1780 nm/RIU

Weak

[14]

SPR

Laser beam modulation engraving

2896.4 nm/RIU

Weak

[15]

TFBG-SPR

Phase mask

1023 nm/RIU

Strong

[22]

LPG-SPR

Laser beam modulation engraving

1600 nm/RIU

Strong

[23]

SPR

Hot melt torsion

3391.15 nm/RIU

Strong

This work

 In addition, in this paper, different types of twisted fiber SPR sensing probes were optimized to obtain experimental probes (step-multimode twisted probe with pitch of 130μm) with higher refractive index sensing sensitivity. The results showed cladding-type SPR sensors, based on step-multimode twisted fibers, excited a higher order cladding mode. The sensors were more sensitive to external environmental refractive index changes and more easily identified the refractive index of the external environment of the gold film due to the specific binding of antibodies to GDF11 antigen. The method realizes the accurate detection of GDF11 solution concentration.

In the surface functionalization of step index multimode twisted fiber SPR sensing probes, the modification of ZIF-67 layer outside the gold film can effectively increase the sensitivity of the fiber sensor. It is due to its unique properties: under the excitation of transmitted light in the fiber, the electrons in ZIF-67 are continuously transferred to the gold film, resulting in enhanced electric field coupling strength on the surface of the gold film, thus increasing the sensitivity of the sensor. The subsequent modified SPA layer, which has the property of binding to the crystallizable segment (Fc) of GDF11 antibody, can realize the function of immobilizing GDF11 antibody and providing sites for specific binding of GDF11 antigen. After occupying the non-specific binding sites on the sensor surface by using BSA solution, it prevents SPR resonance valley movement with the change of the refractive index of the sensor surface due to the non-specific binding sites in the experiment. This is one of the reasons why the sensor can achieve specific detection of GDF11 antigen antibody. In order to show the performance of the sensor more intuitively, the performance of different biochemical sensors is compared in Table 2. It can be seen that the twisted SPR sensor proposed in this paper had a large detection range and a low detection limit, indicating that the sensor has the potential to continue to promote the research of GDF11.

Table-2. Performance comparison of different optical fiber biochemical sensors.

Sensing principle

Detection substances

Detection range

Detection limitation

Reference

SPR

IgG

2 mg/mL-100 mg/mL

0.90 μg/mL

[24]

SPR

MMP-9

10 ng/mL-200 ng/mL

8 pg/mL

[25]

SPR

DNA

10 pM-100 pM

10 pM

[26]

TFBG-SPR

Hg2+

10 pM-1mM

3.073 pM

[16]

LPG-SPR

As3+

0-0.2 ppb

0.04 ppb

[17]

LSPR

DNA

100 pM-1 μM

67 pM

[27]

SPR

GDF11

1 pg/mL-10 ng/mL

0.34 pg/mL

This work

  1. Conclusion

In this paper, a twisted fiber based on cladding-type SPR sensor is proposed to achieve high accuracy detection of GDF11 concentration. The twisted SPR sensors on different types of optical fibers are constructed by using hot melt torsion technique. After the optimization of the fiber type, the average sensitivity of the step-index multimode twisted SPR sensor for refractive index detection reaches 3391.15 nm/RIU. After the functionalization of the GDF11 antibody on the sensor surface, the specific detection of GDF11 antigen is achieved with the detection sensitivity of 2.518 nm/lgC and the detection limit of 0.34 pg/ml. It is potential for this sensor to play an important role in the study of GDF11 mechanism. It is expected to enable specific concentration detection of other members of the growth differentiation factor family by targeted modification of the probe surface with different antibodies. The diameter of the probe is only 125 μm, which realizes the detection of microscopic solutions, as well as access to small spaces such as blood vessels for online detection work.

Author Contributions: Y.W. conceived and designed the experiments; Z.R. and R.W. performed the experiments; Z.R. and C.L. contributed reagents/materials/analysis tools; C.S. and C.W. analyzed the data; Y.Z. wrote the paper. All authors have read and agreed to the published version of the manuscript.

Funding: National Natural Science Foundation of China (No. 61705025); and partially supported by the following grants: Chongqing Natural Science Foundation (cstc2019jcyj-msxmX0607,cstc2019jcyj-msxmX0431), the Science and Technology Project Affiliated to the Education Department of Chongqing Municipality(No. KJZD-M202201201), Chongqing Key Laboratory of Geological Environment Monitoring and Disaster Early-Warning in Three Gorges Reservoir Area (ZD2020A0103, ZD2020A0102), Fundamental Research Funds for Chongqing Three Gorges University of China(No. 19ZDPY08),Open Project Program of Chongqing Key Laboratory of Development and Utilization of Genuine Medicinal Materials in Three Gorges Reservoir Area (No.KFKT2022005), Chongqing Postgraduate Research and Innovation Project.

Data Availability Statement: Not applicable.

Conflicts of Interest: The authors declare no conflict of interest.

References

[1]Lu, Q.; Tu, M.L.; Li, C.J.; Jiang, T.J.; Liu, T.; Luo,X.H. GDF11 Inhibits Bone Formation by Activating Smad2/3 in Bone Marrow Mesenchymal Stem Cells. Calcif. Tissue. Int. 2016, 99, 500–509.

[2]Schafer, M.J.; Lebrasseur, N.K. The influence of GDF11 on brain fate and function. GeroScience. 2019, 41, 1–11.

[3]Loffredo, F.S.; Steinhauser, M.L.; Jay, S.M.; Gannon, J.; Pancoast, J.R.; Yalamanchi, P.; Sinha, M.; Dall’Osso, C.; Khong,D.; Shadrach, J. L. Growth Differentiation Factor 11 Is a Circulating Factor that Reverses Age-Related Cardiac Hypertrophy. Cell. 2013, 153, 828-839.

[4]Egerman, M.A.; Cadena, S.M.; Gilbert, J.A.; Meyer, A.; Nelson, H.N.; Swalley, S.E.; Mallozzi, C.; Jacobi ,C.; L. Jennings, L.; Clay, I. GDF11 Increases with Age and Inhibits Skeletal Muscle Regeneration. Cell. Metab. 2015, 22, 164-174.

[5]Zhang, Y.H.; Wei, Y.; Liu, D.; Liu, F.; Li, X.S.; Pan, L.H.; Pang, Y.; Chen,D.L. Role of growth differentiation factor 11 in development , physiology and disease. Oncotarget. 2017, 8, 81604-81616.

[6]Hosseini, S.; Vázquez-Villegas, P.; Rito-Palomares, M.; Martinez-Chapa, S.O. [SpringerBriefs in Applied Sciences and Technology] Enzyme-linked Immunosorbent Assay (ELISA) || Advantages, Disadvantages and Modifications of Conventional ELISA. Springer. 2018, 5, 67-115.

[7]Liu, G.; Zhao, J.; Wang, S.; Lu, S. Yang, X. Enzyme-induced in situ generation of polymer carbon dots for fluorescence immunoassay. Sens. Actuators. B. Chem. 2020, 306, 127583-127591.

[8 ]An, N.; Li, K.; Zhang, Y.; Wen,T.; Jin, W. A multiplex and regenerable surface plasmon resonance (MR-SPR) biosensor for DNA detection of genetically modified organisms. Talanta. 2021, 231, 122361-122368.

[9]Dong, J.L.; Zhang, Y.X.; Wang, Y.J.; Yang, F.; Hu,S. Side-polished few-mode fiber based surface plasmon resonance biosensor. Opt. Express. 2019, 27, 11348-11360.

[10]Ravindran, N.; Kumar, S.; Yashini, M.; Rajeshwari, S.; Mamathi, C.A.; Nirmal, T.S.; Sunil, C.K. Recent advances in Surface Plasmon Resonance (SPR) biosensors for food analysis, a review. Crit. Rev. Food. Sci. Nutr. 2021, 1-23.

[11]Hayashi, Y.; Mikawa, S.; Masumoto, K.; Katou, F.; Sato, K. GDF11 expression in the adult rat central nervous system. J. Chem. Neuroanat. 2018, 89, 21-36.

[12]Zhang, Y.H.; Pan, L.H.; Pang, Y.; Yang, J.X.; Lv, M.J.; Liu, F.; Chen, X.X.; Gong, H.J.; Liu, D. GDF11/BMP11 as a novel tumor marker for liver cancer. Exp. Ther. Med. 2018, 15,3495-3500.

[13]Esteban, O.; Naranjo, F.B.; Diaz-Herrera, N.; Valdueza-Felip, S.; María-Cruz, N.; González-Canobet, A. High-sensitive SPR sensing with Indium Nitride as a dielectric overlay of optical fibers. Sens. Actuators. B. Chem2011, 158(1): 372-376.

[14] Cennamo, N.; Arcadio, F.; Zeni, L.; Catalano, E.; Del Prete, D.; Buonanno, G.; Minardo, A. The Role of Tapered Light-Diffusing Fibers in Plasmonic Sensor Configurations. Sensors (Basel). 2021, 21(19).

[15] Wei, Y.; Li, L.; Liu, C.; Wang, R.; Zhao, X.; Ran, Z.; Jiang, T. High sensitivity fiber cladding SPR strain sensor based on V-groove structure. Opt Express. 2022, 30(5): 7412-7425.

[16] Duan, Y.; Wang, F.; Zhang, X.; Liu, Q.; Lu, M.; Ji, W.; Zhang, Y.; Jing, Z.; Peng, W. TFBG-SPR DNA-Biosensor for Renewable Ultra-Trace Detection of Mercury Ions. Journal of Lightwave Technology. 2021, 39(12): 3903-3910.

[17] Huang, C.; Zhou, Y.; Yu, G.; Zeng, J.; Li, Q.; Shen, K.; Wu, X.; Guo, R.; Zhang, C.; Zheng, B.; Wang, J. Glutathione-functionalized long-period fiber gratings sensor based on surface plasmon resonance for detection of As(3+) ions. Nanotechnology. 2021, 32(48).

[18]Jiang, C.; Liu, Y.; Huang, L.; Mou, C. Double Cladding Fiber Chiral Long-Period Grating-Based Directional Torsion Sensor. IEEE. Photonics. Technol. Lett. 2019, 31, 1522-1525.

[19]Liu, Y.; Deng, H.; Yuan, L. Directional torsion and strain discrimination based on Mach-Zehnder interferometer with off-axis twisted deformations. Opt. Laser. Technol. 2019, 120, 105754-105761.

[20]Gahlaut, S.K.; Pathak, A.; Gupta, B.D.; Singh, J.P. Portable fiber-optic SPR platform for the detection of NS1-antigen for dengue diagnosis. Biosens. Bioelectron. 2022, 196, 113720-113728.

[21]Wang, Q.; Jing, J.Y.; Wang, B.T. Highly Sensitive SPR Biosensor Based on Graphene Oxide and Staphylococcal Protein A Co-Modified TFBG for Human IgG Detection. IEEE. Trans. Instrum. Meas. 2018, 68, 3350-3357.

[22]Pham, X.; Si, J.; Chen, T.; Chah, K.; Zubia, J.; Villatoro, J.; Caucheteur, C. Demodulation method for tilted fiber Bragg grating refractometer with high sensitivity. Journal of Applied Physics2018, 123(17).

[23]Hu, H.F.; Deng, Z.Q.; Zhao, Y.; Li, J.; Wang, Q. Sensing Properties of Long Period Fiber Grating Coated by Silver Film. IEEE Photonics Technology Letters2015, 27(1): 46-49.

[24]Shi, S.; Wang, L.; Su, R.; Liu, B.; Huang, R.; Qi, W.; He, Z. A polydopamine-modified optical fiber SPR biosensor using electroless-plated gold films for immunoassays. Biosens Bioelectron. 2015, 74: 454-60.

[25]Mohseni, S.; Moghadam, T.T.; Dabirmanesh, B.; Jabbari, S.; Khajeh, K. Development of a label-free SPR sensor for detection of matrixmetalloproteinase-9 by antibody immobilization on carboxymethyldextran chip. Biosensors and Bioelectronics. 2016, 81: 510-516.

[26]Li, C.; Gao, J.; Shafi, M.; Liu, R.; Zha, Z.; Feng, D.; Liu, M.; Du, X.; Yue, W.; Jiang, S. Optical fiber SPR biosensor complying with a 3D composite hyperbolic metamaterial and a graphene film. Photonics Research. 2021, 9(3).

[27]Lu, M.; Peng, W.; Lin, M.; Wang, F.; Zhang, Y. Gold Nanoparticle-Enhanced Detection of DNA Hybridization by a Block Copolymer-Templating Fiber-Optic Localized Surface Plasmon Resonance Biosensor. Nanomaterials (Basel). 2021, 11(3).

Reviewer 2 Report

In the manuscript Twisted fiber optic SPR sensor for GDF11 concentration detection, the authors demonstrated three SPR sensors based on different twisted fibers.

There are a few questions related to this work.

1: I am afraid this is not a label-free sensor since the sensor cannot directly detect GDF-11 with specificity. 

2: Comparison should be provided to demonstrate the improvements of the sensor.

I think this work could be important to specific readers, and the authors should emphasize the novelty and improvements.

A few places have typos.

Author Response

Reviewer 2: 

In the manuscript “Twisted fiber optic SPR sensor for GDF11 concentration detection”, the authors demonstrated three SPR sensors based on different twisted fibers.There are a few questions related to this work.

  1. I am afraid this is not a label-free sensor since the sensor cannot directly detect GDF-11 with specificity. 

Answer: Thanks for the suggestion. The sensor is a label-free sensor, because the basis for determining whether the sensor is a label sensor is whether the object to be tested is marked. In the whole process of the sensor sensing test, we did not label the analyte, and the modified material only played a role in stimulating the SPR effect, increasing the sensitivity of the sensor, and adsorbing the antibody molecule. In the common biochemical detection method with labeling, ELISA requires the use of enzyme molecules. Combined with the antibody molecule, the antibody molecule is labeled. After the labeling is completed, it is combined with the antigen to be tested. After the antigen antibody is fully combined, the substrate solution is added drop by drop to react with the labeled enzyme molecule, and the biochemical detection is realized by analyzing the chromogenic reaction. The principle of fluorescence immunoassay is to directly use fluorescent materials to label antibody molecules and combine with unlabeled antigen molecules. After the antigen and antibody are fully combined, fluorescence photos are taken by fluorescence microscope, so as to judge the binding of antigen and antibody by observing and analyzing the fluorescent materials of markers. The sensor does not directly and specifically detect GDF11, but needs to characterize the refractive index change of the outer surface of the sensor through the movement of the resonance wavelength to specifically detect GDF11.

  1. Comparison should be provided to demonstrate the improvements of the sensor.

Answer: Thanks for the suggestion. We have supplemented the Table 1 in the paper. Compared with other cladding sensors, the proposed sensor has some improvements in physical strength and detection sensitivity, as shown in Table 1.

We have added the above content into the discussion, as follows :

In this paper, we used optical fiber hot melt twisting technology to effectively couple the light from the core into the cladding. We constructed a new optical fiber cladding SPR sensor with fast and simple processing, and the fiber structure was not damaged and still maintained strong physical strength [18,19]. In table 1, we compared the sensing performances of different types of cladding sensors. It can be seen that the torsional SPR sensor proposed in this paper had higher physical strength and higher sensing sensitivity. The optical fibers as biosensing elements can enter into narrow spaces such as blood vessels for detection work, such as protein concentration detection in blood environment [20,21].

Table-1. Performance comparison of different cladding SPR sensors.

Sensing principle

Processing method

Refractive index sensing performance

Physical strength

Reference

SPR

Taper

1780 nm/RIU

Weak

[14]

SPR

Laser beam modulation engraving

2896.4 nm/RIU

Weak

[15]

TFBG-SPR

Phase mask

1023 nm/RIU

Strong

[22]

LPG-SPR

Laser beam modulation engraving

1600 nm/RIU

Strong

[23]

SPR

Hot melt torsion

3391.15 nm/RIU

Strong

This work

3.I think this work could be important to specific readers, and the authors should emphasize the novelty and improvements.

Answer: Thanks for the suggestion. Due to the confinement of the cladding to the core light field, the SPR sensing structure cannot be realized. In order to make the light in the core can contact the gold film to construct the SPR sensing structure, it is often necessary to micro-process the fiber to remove the cladding or couple the light in the core to the cladding, such as etching, tapering, laser etching, etc. The processing process is complex and the fiber probe becomes fragile. In addition, tilted FBG and LPG can also effectively couple the light in the fiber core to the fiber cladding to form a grating-structured cladding SPR sensor. These two types of grating SPR sensors are fabricated on the fiber core without damaging the overall structure of the fiber, and have the advantages of high physical strength and good repeatability. However, compared with conventional optical fiber SPR sensor, TFBG-SPR and LPG-SPR sensors have the disadvantages of low sensitivity and complicated fabrication process. Compared with other cladding SPR sensors, this paper uses hot melt torsion technology to prepare torsion structure, realizes SPR sensing without affecting the overall structure of optical fiber, and constructs a new cladding sensor. In addition, the sensor realizes the specific detection of low concentration GDF11, which is helpful for the study of GDF11 mechanism. In terms of performance improvement, compared with other cladding SPR sensors, the cladding sensor proposed in this paper has a simple production method, and the physical strength and refractive index sensing performance have been improved. We have added the above content to the discussion part.

4.A few places have typos.

Answer: Thanks for the suggestion. We have modified the spelling mistakes in the article. For example, the"coarse core" was replaced as"large core diameter".We deleted the extra word "figure" in Section 2.2.

Reviewer 3 Report

Dear Authors,

the article is very interesting and well-written. Nevertheless, there are some issues.

1)      The number of references is too small;

2)      Please cite similar attempts in your field of research achieved by other groups and compare the detection limits;

3)      In Fig. 7 (b) the experimental data are reported without the error bars; Why? Did you repeat the measurements or not?

4)      What about the stability of the system? How many cycles of measurement can you perform with this device?

5)      Have you patented this device?

Author Response

Reviewer 3:

the article is very interesting and well-written. Nevertheless, there are some issues. 1.The number of references is too small;

Answer: Thanks for the suggestion. We have supplemented the references in the paper.

[1]Lu, Q.; Tu, M.L.; Li, C.J.; Jiang, T.J.; Liu, T.; Luo,X.H. GDF11 Inhibits Bone Formation by Activating Smad2/3 in Bone Marrow Mesenchymal Stem Cells. Calcif. Tissue. Int. 2016, 99, 500–509.

[2]Schafer, M.J.; Lebrasseur, N.K. The influence of GDF11 on brain fate and function. GeroScience. 2019, 41, 1–11.

[3]Loffredo, F.S.; Steinhauser, M.L.; Jay, S.M.; Gannon, J.; Pancoast, J.R.; Yalamanchi, P.; Sinha, M.; Dall’Osso, C.; Khong,D.; Shadrach, J. L. Growth Differentiation Factor 11 Is a Circulating Factor that Reverses Age-Related Cardiac Hypertrophy. Cell. 2013, 153, 828-839.

[4]Egerman, M.A.; Cadena, S.M.; Gilbert, J.A.; Meyer, A.; Nelson, H.N.; Swalley, S.E.; Mallozzi, C.; Jacobi ,C.; L. Jennings, L.; Clay, I. GDF11 Increases with Age and Inhibits Skeletal Muscle Regeneration. Cell. Metab. 2015, 22, 164-174.

[5]Zhang, Y.H.; Wei, Y.; Liu, D.; Liu, F.; Li, X.S.; Pan, L.H.; Pang, Y.; Chen,D.L. Role of growth differentiation factor 11 in development , physiology and disease. Oncotarget. 2017, 8, 81604-81616.

[6]Hosseini, S.; Vázquez-Villegas, P.; Rito-Palomares, M.; Martinez-Chapa, S.O. [SpringerBriefs in Applied Sciences and Technology] Enzyme-linked Immunosorbent Assay (ELISA) || Advantages, Disadvantages and Modifications of Conventional ELISA. Springer. 2018, 5, 67-115.

[7]Liu, G.; Zhao, J.; Wang, S.; Lu, S. Yang, X. Enzyme-induced in situ generation of polymer carbon dots for fluorescence immunoassay. Sens. Actuators. B. Chem. 2020, 306, 127583-127591.

[8 ]An, N.; Li, K.; Zhang, Y.; Wen,T.; Jin, W. A multiplex and regenerable surface plasmon resonance (MR-SPR) biosensor for DNA detection of genetically modified organisms. Talanta. 2021, 231, 122361-122368.

[9]Dong, J.L.; Zhang, Y.X.; Wang, Y.J.; Yang, F.; Hu,S. Side-polished few-mode fiber based surface plasmon resonance biosensor. Opt. Express. 2019, 27, 11348-11360.

[10]Ravindran, N.; Kumar, S.; Yashini, M.; Rajeshwari, S.; Mamathi, C.A.; Nirmal, T.S.; Sunil, C.K. Recent advances in Surface Plasmon Resonance (SPR) biosensors for food analysis, a review. Crit. Rev. Food. Sci. Nutr. 2021, 1-23.

[11]Hayashi, Y.; Mikawa, S.; Masumoto, K.; Katou, F.; Sato, K. GDF11 expression in the adult rat central nervous system. J. Chem. Neuroanat. 2018, 89, 21-36.

[12]Zhang, Y.H.; Pan, L.H.; Pang, Y.; Yang, J.X.; Lv, M.J.; Liu, F.; Chen, X.X.; Gong, H.J.; Liu, D. GDF11/BMP11 as a novel tumor marker for liver cancer. Exp. Ther. Med. 2018, 15,3495-3500.

[13]Esteban, O.; Naranjo, F.B.; Diaz-Herrera, N.; Valdueza-Felip, S.; María-Cruz, N.; González-Canobet, A. High-sensitive SPR sensing with Indium Nitride as a dielectric overlay of optical fibers. Sens. Actuators. B. Chem2011, 158(1): 372-376.

[14] Cennamo, N.; Arcadio, F.; Zeni, L.; Catalano, E.; Del Prete, D.; Buonanno, G.; Minardo, A. The Role of Tapered Light-Diffusing Fibers in Plasmonic Sensor Configurations. Sensors (Basel). 2021, 21(19).

[15] Wei, Y.; Li, L.; Liu, C.; Wang, R.; Zhao, X.; Ran, Z.; Jiang, T. High sensitivity fiber cladding SPR strain sensor based on V-groove structure. Opt Express. 2022, 30(5): 7412-7425.

[16] Duan, Y.; Wang, F.; Zhang, X.; Liu, Q.; Lu, M.; Ji, W.; Zhang, Y.; Jing, Z.; Peng, W. TFBG-SPR DNA-Biosensor for Renewable Ultra-Trace Detection of Mercury Ions. Journal of Lightwave Technology. 2021, 39(12): 3903-3910.

[17] Huang, C.; Zhou, Y.; Yu, G.; Zeng, J.; Li, Q.; Shen, K.; Wu, X.; Guo, R.; Zhang, C.; Zheng, B.; Wang, J. Glutathione-functionalized long-period fiber gratings sensor based on surface plasmon resonance for detection of As(3+) ions. Nanotechnology. 2021, 32(48).

[18]Jiang, C.; Liu, Y.; Huang, L.; Mou, C. Double Cladding Fiber Chiral Long-Period Grating-Based Directional Torsion Sensor. IEEE. Photonics. Technol. Lett. 2019, 31, 1522-1525.

[19]Liu, Y.; Deng, H.; Yuan, L. Directional torsion and strain discrimination based on Mach-Zehnder interferometer with off-axis twisted deformations. Opt. Laser. Technol. 2019, 120, 105754-105761.

[20]Gahlaut, S.K.; Pathak, A.; Gupta, B.D.; Singh, J.P. Portable fiber-optic SPR platform for the detection of NS1-antigen for dengue diagnosis. Biosens. Bioelectron. 2022, 196, 113720-113728.

[21]Wang, Q.; Jing, J.Y.; Wang, B.T. Highly Sensitive SPR Biosensor Based on Graphene Oxide and Staphylococcal Protein A Co-Modified TFBG for Human IgG Detection. IEEE. Trans. Instrum. Meas. 2018, 68, 3350-3357.

[22]Pham, X.; Si, J.; Chen, T.; Chah, K.; Zubia, J.; Villatoro, J.; Caucheteur, C. Demodulation method for tilted fiber Bragg grating refractometer with high sensitivity. Journal of Applied Physics2018, 123(17).

[23]Hu, H.F.; Deng, Z.Q.; Zhao, Y.; Li, J.; Wang, Q. Sensing Properties of Long Period Fiber Grating Coated by Silver Film. IEEE Photonics Technology Letters2015, 27(1): 46-49.

[24]Shi, S.; Wang, L.; Su, R.; Liu, B.; Huang, R.; Qi, W.; He, Z. A polydopamine-modified optical fiber SPR biosensor using electroless-plated gold films for immunoassays. Biosens Bioelectron. 2015, 74: 454-60.

[25]Mohseni, S.; Moghadam, T.T.; Dabirmanesh, B.; Jabbari, S.; Khajeh, K. Development of a label-free SPR sensor for detection of matrixmetalloproteinase-9 by antibody immobilization on carboxymethyldextran chip. Biosensors and Bioelectronics. 2016, 81: 510-516.

[26]Li, C.; Gao, J.; Shafi, M.; Liu, R.; Zha, Z.; Feng, D.; Liu, M.; Du, X.; Yue, W.; Jiang, S. Optical fiber SPR biosensor complying with a 3D composite hyperbolic metamaterial and a graphene film. Photonics Research. 2021, 9(3).

[27]Lu, M.; Peng, W.; Lin, M.; Wang, F.; Zhang, Y. Gold Nanoparticle-Enhanced Detection of DNA Hybridization by a Block Copolymer-Templating Fiber-Optic Localized Surface Plasmon Resonance Biosensor. Nanomaterials (Basel). 2021, 11(3).

2.Please cite similar attempts in your field of research achieved by other groups and compare the detection limits;

Thanks for the suggestion. We have supplemented the Table 2 in the paper. In the field of optical fiber biochemical sensing, the work of this paper has a wider detection range and a lower detection limit than other similar works, as shown in Table 2.

Table-2. Performance comparison of different optical fiber biochemical sensors.

Sensing principle

Detection substances

Detection range

Detection limitation

Reference

SPR

IgG

2 mg/mL-100 mg/mL

0.90 μg/mL

[24]

SPR

MMP-9

10 ng/mL-200 ng/mL

8 pg/mL

[25]

SPR

DNA

10 pM-100 pM

10 pM

[26]

TFBG-SPR

Hg2+

10 pM-1mM

3.073 pM

[16]

LPG-SPR

As3+

0-0.2 ppb

0.04 ppb

[17]

LSPR

DNA

100 pM-1 μM

67 pM

[27]

SPR

GDF11

1 pg/mL-10 ng/mL

0.34 pg/mL

This work

3.In Fig. 7 (b) the experimental data are reported without the error bars; Why? Did you repeat the measurements or not?

Thanks for the suggestion. We have revised the Figure 7(b) in the paper.During the experiment, we performed three repeated measurements on GDF11, and the modified Figure 7 ( b ) is shown as follows :

Figure 7 ( b )

4.What about the stability of the system? How many cycles of measurement can you perform with this device?

Thanks for the suggestion. The stability of the system is good, the device can perform at least 3 measurement cycles. After each measurement, the sensing area was rinsed with NaOH solution at a concentration of 10 mM, which could destroy the binding between GDF11 antigen and antibody, and then the NaOH residue and the detached antigen molecules were rinsed with PBS buffer solution. Multiple strong alkali solution washing will affect the combination of the inner layer of the sensor modified material, reducing the detection performance of the sensor, the experiment found that after 3 measurement cycles, the sensor performance began to decline.

5.Have you patented this device?

Thanks for the suggestion. I have filed a patent application for this device.
